# Magnetic and Resonance Properties of a Low-Dimensional Cobalt–Aluminum Oxide–Germanium Film Tunnel Junction Deposited by Magnetron Sputtering

**Aleksandr V. Kobyakov** [1,2,*], **Gennadiy S. Patrin** [1,2], **Vasiliy I. Yushkov** [1,2], **Yaroslav G. Shiyan** [1,2], **Roman Yu. Rudenko** [1], **Nikolay N. Kosyrev** [2,3] **and Sergey M. Zharkov** [1,2]

1 Institute of Engineering Physics and Radio Electronics, Siberian Federal University, 660041 Krasnoyarsk, Russia
2 Kirensky Institute of Physics, Krasnoyarsk Scientific Center, Siberian Branch, Russian Academy of Sciences, 660041 Krasnoyarsk, Russia
3 Achinsk Branch of Krasnoyarsk State Agrarian University, 662155 Achinsk, Russia
* Correspondence: nanonauka@mail.ru

**Abstract:** The temperature behavior of saturation magnetization and the temperature behavior of the integral signal intensity in electron magnetic resonance spectra is experimentally studied comprehensively using a low-dimensional $Al_2O_3/Ge/Al_2O_3/Co$ (aluminum oxide–cobalt–aluminum oxide–germanium) tunnel junction with different deposition velocities of a ferromagnetic metal (Co) thin layer and non-magnetic layers ($Al_2O_3/Ge/Al_2O_3$). The cobalt ferromagnetic layer was deposited on aluminum oxide in two ways: in one cycle of creating the structure and with atmospheric injection before deposition of the cobalt layer. The thermomagnetic curves revealed the appearance of minima observed at low temperatures on both sides of the cobalt layer. Possible sources of precession perturbations at low temperatures can be explained by: the influence of the $Al_2O_3$ layer structure on the $Al_2O_3/Co$ interface; residual gases in the working chamber atmosphere and finely dispersed cobalt pellets distributed over the cobalt film thickness. The work offers information of great significance in terms of practical application, for both fundamental physics and potential applications of ultrathin films.

**Keywords:** ferromagnetic film; anisotropy tunnel contact; aluminum oxides

## 1. Introduction

Tunneling transition in thin dielectric oxide layers in contact with a ferromagnetic source of spin-polarized electrons has been extensively studied in the past two decades [1–3].

The discovered phenomena of resonant tunneling, which reduces thermal losses [4,5], tunnel spin-galvanic effect [6], spin scattering at the interface [1], and other effects makes tunnel barriers and interfaces an intriguing field of forefront research. Because of their interesting physical properties, including with efficient spin injection, and reduction of heat losses, these contact oxide layer/ferromagnetic metal approaches can be used in various fields, such as for elements of optoelectronic devices, microwave devices, and ultrafast microelectronic devices, including magnetoresistive RAM (MRAM), high-precision, heat-resistant, and high-speed spin sensors [7–10], and in automotive units such as the ABS anti-lock braking system [11].

The problems of transport of spin-polarized carriers in hybrid structures with tunnel junctions are multifaceted. There are fundamental problems in solid-state physics related to the influence of structural, dimensional, and interface effects on the magnetic characteristics of thin-film structures and on spin-dependent transport. Ferromagnetic contacts should be homogeneously magnetized and not contain domains with different magnetization directions. Injection of spin-polarized electrons from such electrodes can

lead to partial and even complete compensation of the integral magnetization of the electron gas in the dielectric/semiconductor. In metal/semiconductor contacts, spin scattering appears at the interface itself [1]. Therefore, the interface must not contain so-called "dead" areas—disordered layers or defects, for example, in the form of punctures in the layers, caused by previous layers. There appear to be difficulties in effective spin current injection associated with the problem of suppression of the formed Schottky barrier, which causes a mismatch of conductivities between the metal layer and the semiconductor [12–14].

Thus, the implementation of spin injection in real-world structures faces both the limited set of materials that allow spin transfer and the technological difficulties in their creation and the technological compatibility of ferromagnetic metals, tunnel barriers, and semiconductors. However, the electrical transport of oxides can be controlled by tuning the interface oxide/ferromagnetic metal.

Oxide tunnel barriers at the 3d metal/semiconductor interface can provide a solution to the conductivity mismatch problem where the main problem is to control the interface between the dielectric and the device electrodes [15]. Among oxide tunnel barriers, $Al_2O_3$ shows itself as a high quality atomic-thin dielectric. This material shows high barrier height, low tunneling current density, and high breakdown field strength [16–18]. Superparamagnetic Co-Al-O films have giant magnetoresistance due to spin-dependent tunneling between granules [19]. The distinguishing features of cobalt properties from other 3d metals are: high saturation magnetization (MS), high Curie temperature (TC), rather high degree of polarization of conduction electrons [20,21]. It is known, however, that the conditions of cobalt synthesis affect the phase formation leading to single, or more often mixed phases in the formed cobalt layer. However, rational control in the synthesis of cobalt (II) complexes with the desired magnetic behavior remains a major challenge, and much more work is needed to expand our knowledge of the structure-property relationships [22–24].

Among the methods for studying the magnetic and resonant properties of ferromagnetic metal particles or clusters, one can single out methods for combining measurements of magnetization and measurements of electron paramagnetic resonance. For example, such methods have yielded results in the study of the magnetoresistive and magnetocaloric effect in $La_{1-x}Ba_xMnO_3$ compounds for explaining the role of the Griffiths phase with ferromagnetic metal clusters above Curie temperature [25].

Earlier, in [26], the temperature dependence of the coercivity was analyzed in the $Al_2O_3/Ge/Al_2O_3/Co$ structure at different sputtering rates of the system. However, so far, the role of the cobalt/aluminum oxide contact deposition regime on the physical and chemical properties of materials remains yet to be investigated further. Such a study of magnetic properties and Co-Al-O/semiconductor interfaces in relation to technological synthesis conditions is of notable interest to ensure the transition to the next generation of microelectronics.

In this work, we have investigated the following hitherto unresolved question: what is the behavior of the magnetic properties of the films at different roughnesses and velocities of the magnetic and non-magnetic $Al_2O_3/Ge/Al_2O_3/Co$ layers? The question is addressed by investigating the related physical properties of $Al_2O_3/Ge/Al_2O_3/Co$ layers. The temperature behavior of saturation magnetization and the temperature behavior of the integral signal intensity in the electron magnetic resonance spectra is experimentally studied.

## 2. Materials and Methods

The $Al_2O_3$ (130 nm)/ Ge (45 nm)/ $Al_2O_3$ (4.5 nm)/ Co (95 nm) films were synthesized by ion-plasma sputtering onto a glass substrate at a base pressure of P = $10^{-7}$ Torr in an argon atmosphere at 3 mTorr. The substrate was pre-cleaned by ion-plasma etching in the working chamber, immediately before the sputtering process. Sputtering was carried out on a rotating substrate at its temperature T $\approx$ 373 K.

The difference in the surface roughness values of the cobalt layer was achieved in two ways:

"A" Alternating synthesis of nonmagnetic $Al_2O_3/Ge/Al_2O_3$ layers and magnetic Co layer in one cycle.

"B" Alternating synthesis of $Al_2O_3/Ge/Al_2O_3$ nonmagnetic layers, then pumping air into the system and pumping to the base pressure. Then synthesis of the magnetic cobalt layer.

Four sets of samples were obtained in each sputtering method with different combinations of deposition rates of magnetic and non-magnetic layers. The difference in sputtering rates by a factor of five was controlled by changing the power maintained on the magnetrons. The surface structure of the films was investigated using a Veeco Multi Mode atomic force microscope (1 nm resolution). The structure and thicknesses of the layers were determined on a JEOL JEM-2100 electron microscope (during sample preparation at the Gatan PIPS facility).

The phase composition was investigated by X-ray diffraction (XRD) using a DRON-4-07. The analysis of the intensity of the X-ray diffraction reflections were made using the ICDD PDF 4+ crystallographic database.

Table 1 shows the average values of the deposition rate and roughness parameters (the standard deviation of the roughness profile (rms) and the height of irregularities determined by 10 main points (Rz)) of the cobalt surface.

**Table 1.** Labeling of samples, velocity, roughness.

| | Samples | A1 | B1 | A2 | B2 | A3 | B3 | A4 | B4 |
|---|---|---|---|---|---|---|---|---|---|
| **Deposition Rate, nm/min** | Layer 1: $Al_2O_3$ (130 nm) | | | 0.55 | | | | 0.05 | |
| | Layer 2: Ge (45 nm) | | | 14.4 | | | | 2.4 | |
| | Layer 3: $Al_2O_3$ (4.5 nm) | | | 0.55 | | | | 0.05 | |
| | Layer 4: Co (95 nm) | 7.2 | | 1.2 | | 1.2 | | 7.2 | |
| Average parameters of cobalt surface roughness. | Rms (nm.) | 8.3 | 16.5 | 4 | 12 | 4.4 | 13 | 5.3 | 14.5 |
| | Max height (Rmax) (nm.) | 53 | 133 | 39 | 156 | 48 | 122 | 51 | 182 |
| | Rz (nm.) | 51.5 | 115 | 36 | 96 | 38 | 106 | 47 | 114 |

Magnetization measurements were carried out on the MPMS-XL machine and using the magneto-optical Kerr effect method (NanoMOKE-2). The magnetic field lay in the plane of the film. Before each measurement, the film was first placed in a demagnetizer and then cooled in a zero magnetic field (ZFC mode). Using the magneto-optical Kerr effect (MOKE), the saturation magnetization was measured by the method described in the work of N. N. Kosyrev, V. Yu. Yakovchuk, G. S. Patrin, V. A. Komarov, E. N. Volchenko & I. A. Tarasov // Optical and Magnetic Properties of the $Dy_xCo_{1-x}/Bi/Py$ Trilayers // Technical Physics Letters // V.47, P.P. 107–110 (2021). To measure the resonance properties, a Bruker E 500 CW EPR spectrometer operating at a frequency $f_{MWF} = 9.48$ GHz was used.

*Experimental Results and Discussion*

A cross-section of the film shows the presence of distinct interfaces between all layers (see Figure 1 for sample A1). The analysis of diffraction reflections observed in the selected electron diffraction area (SAED) obtained from sample A1 (Figure 1a) shows that the Ge and $Al_2O_3$ layers have an amorphous structure, and the Co layer has a crystalline structure. Interpretation of electron diffraction reflections (Figure 1d) allows us to conclude that both samples (A1) contain hexagonal densely packed (hcp) and face-centered cubic (fcc) Co phases. The analysis of the averaged intensity profiles of electron diffraction reflections makes it possible to estimate the content of the hcp and fcc Co phases. The obtained fraction is $\approx$ 90% hcp-Co and $\approx$ 10% fcc-Co, and it is almost the same for all samples. The EDS cartographic images (Figure 1b,c) clearly demonstrated the distribution of O and Al elements in the Co layer, which strongly suggests that the Co clusters contain impurities of these elements uniformly throughout the volume.

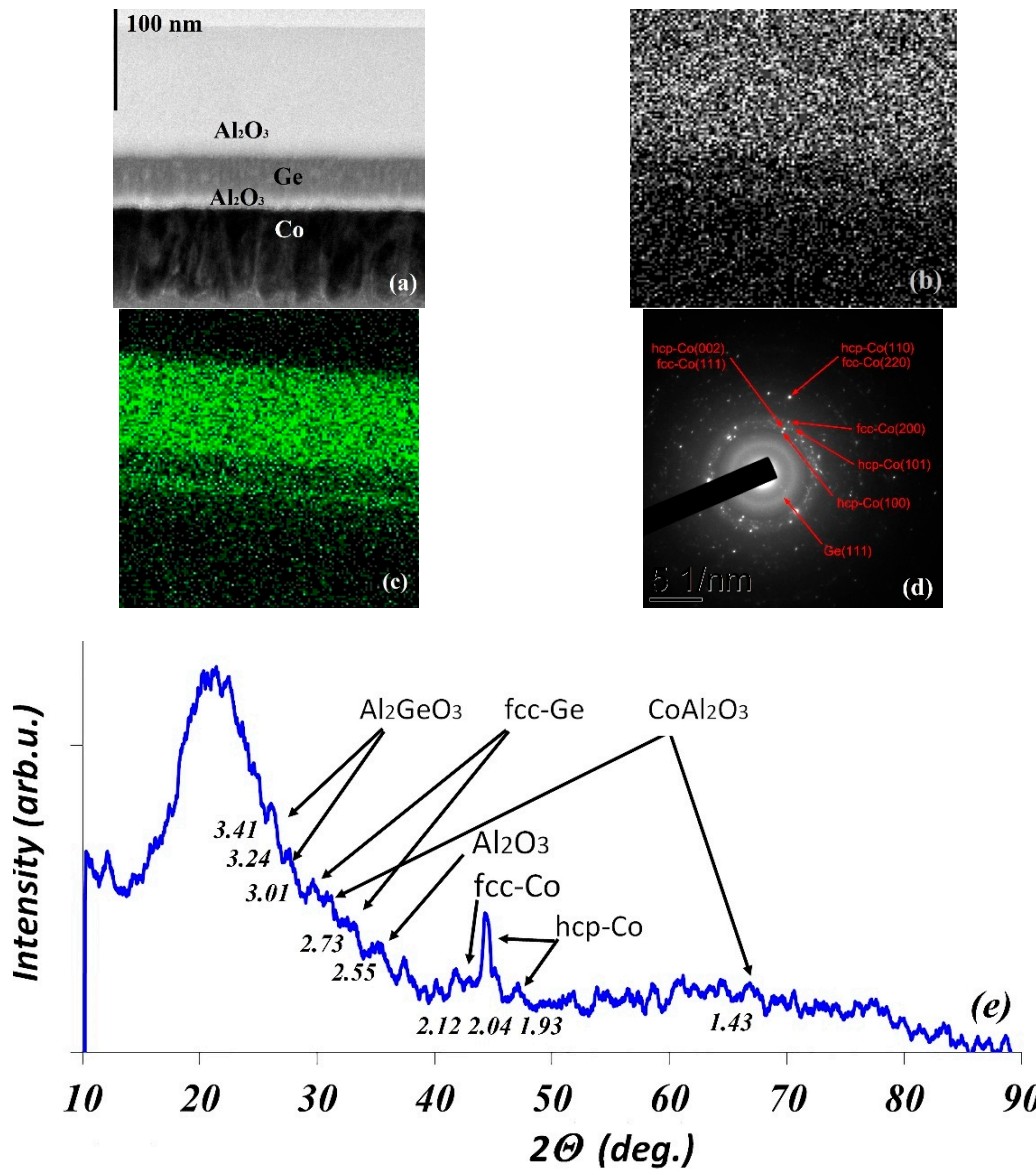

**Figure 1.** STEM image (**a**), the corresponding EDS elemental mapping images of O (**b**) and Al (**c**), corresponding SAED patterns (**d**) and XRD patterns (**e**) of the sample A1—$Al_2O_3$/Ge/$Al_2O_3$/Co.

Figure 1e shows the XRD patterns of the sample A1—$Al_2O_3$/Ge/$Al_2O_3$/Co. Diffraction peaks indicate the presence of hcp- and fcc-Co phases. Germanium is present in the cubic and amorphous phase. Peaks for $Al_2O_3$ and some of its compounds are present.

The SEM micrographs of the samples in this study are shown in Figure 2. It can be seen from particle size distribution histograms that the average particle size of structure $Al_2O_3$/Ge/$Al_2O_3$/Co. of sample A1 is estimated to be 28 nm. The average particle size of sample A2 is estimated to be 30 nm.

Figure 3 shows the FMR response spectra of cobalt for samples A1 and B1. The obtained spectra are well approximated by the superposition of two Lorentzian-type lines, except for the data for sample A2, with a single line. This is due to the weak line intensity.

The dependences of the integral intensities (I, the areas under the absorption lines of resonance spectra) of magnetic resonance spectra on temperature are presented on the right y-axis in Figure 4a–d for samples of series A and in Figure 5a–d for samples of series B. At low temperatures, a minimum of the integral intensities of the resonance spectra appears. The position of the minimum differs for different samples.

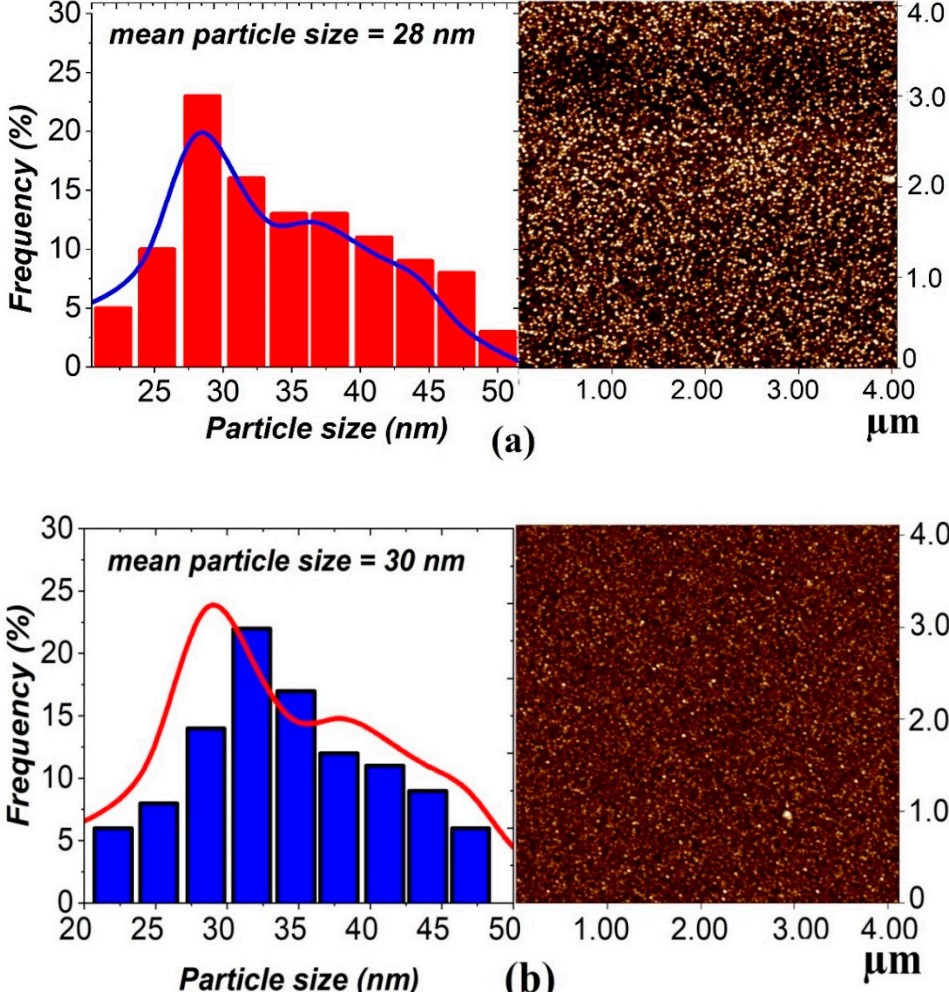

**Figure 2.** The corresponding particle size distribution histograms (left) and SEM images (right) for samples of A1 (**a**) and A3 (**b**) for structure $Al_2O_3/Ge/Al_2O_3/Co$.

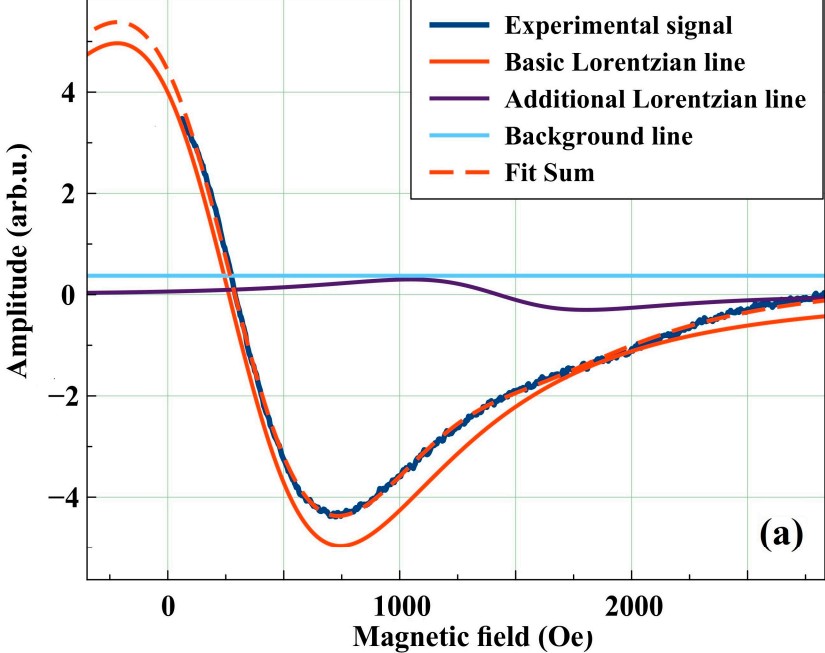

**Figure 3.** *Cont.*

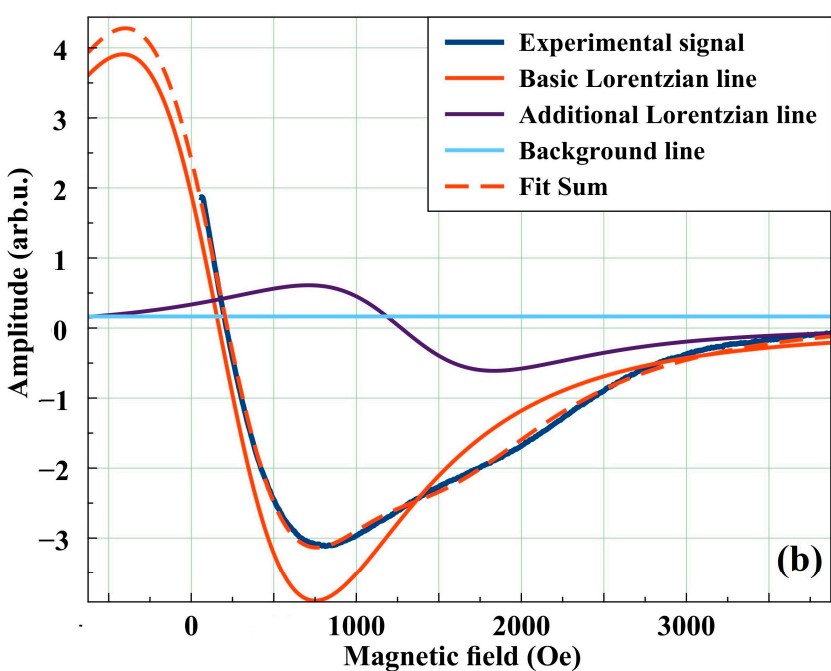

**Figure 3.** Ferromagnetic resonance spectra at 160K for samples A1 (**a**) and B1 (**b**).

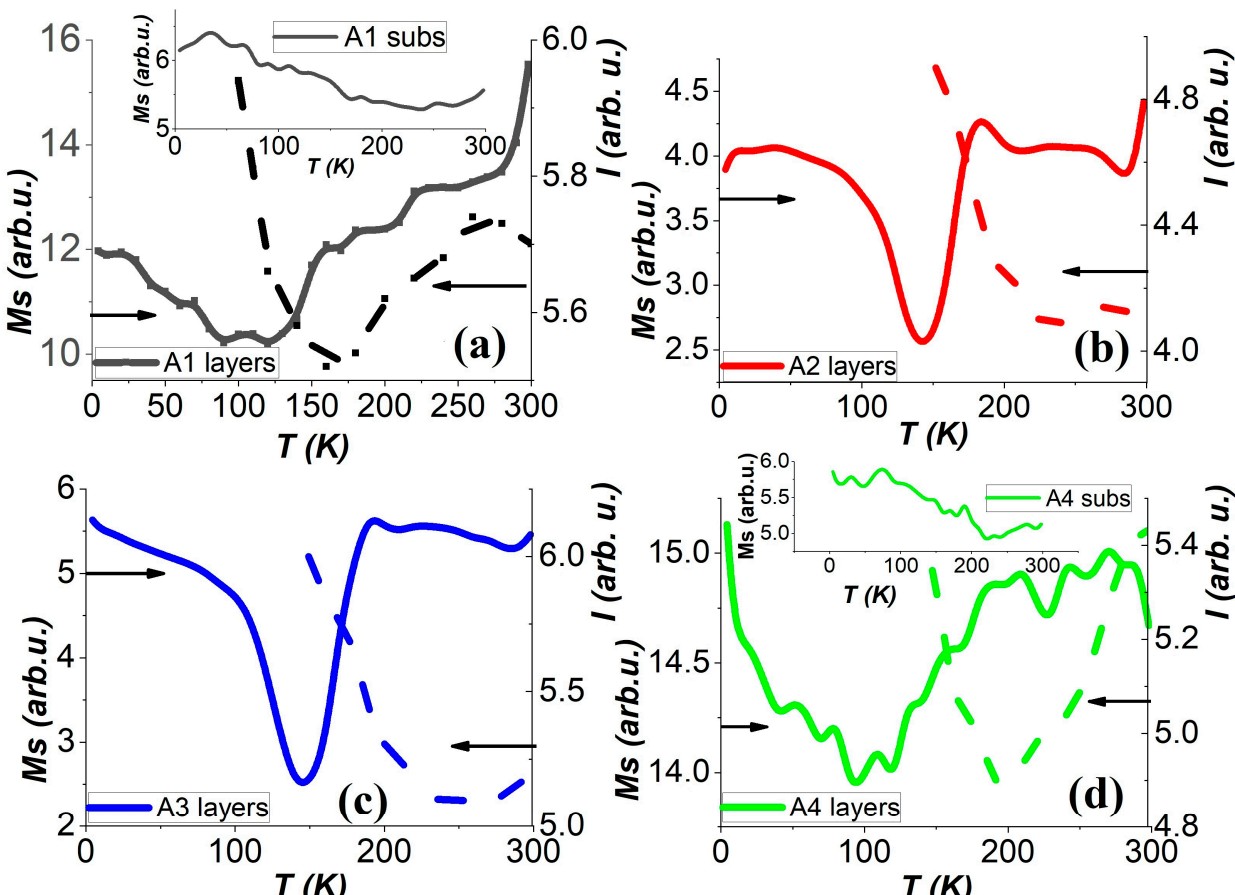

**Figure 4.** The right y-axis is the temperature dependence of the saturation magnetization values ($M_S$) under H = 1.2 kOe measured by the Kerr effect method from the side of the layers of samples A1 (**a**), A2 (**b**), A3 (**c**), A4 (**d**) and from the side of the substrate on the box (for samples A1, A4) for structures $Al_2O_3/Ge/Al_2O_3/Co$. The left y-axis is the temperature dependence of the integral intensities of the FMR spectra of the corresponding samples.

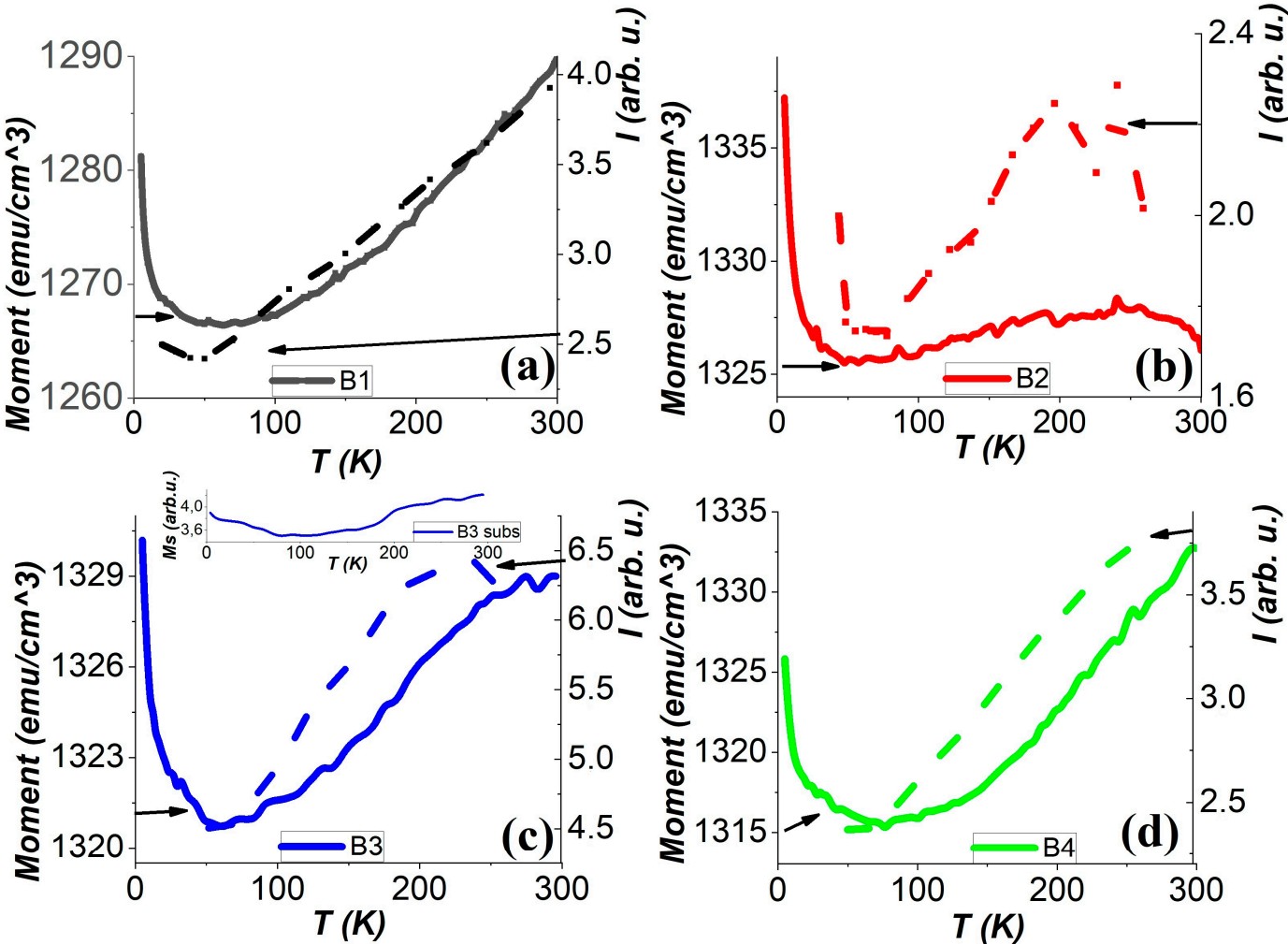

**Figure 5.** The right y-axis is the temperature dependence of the saturation magnetization values ($M_S$) under H = 1.2 kOe measured by SQUID magnetometry of samples B1 (**a**), B2 (**b**), B3 (**c**), B4 (**d**) for structures $Al_2O_3/Ge/Al_2O_3/Co$. The box shows dependence for the 3B sample measured by the Kerr effect method. The left y-axis is the temperature dependence of the intensities of the FMR spectra of the corresponding samples.

The saturation magnetization as a function of temperature $M_S$ (T) for $Al_2O_3/Ge/Al_2O_3/Co$ samples after zero-field cooling (ZFC) condition was measured under a 1.2 kOe field in the temperature range of 4.2–300 K. The dependences $M_S$ (T) for samples of series A are shown on the left y-axis in the Figure 4a–d. The dependence was obtained using Kerr measurements on the front side of the cobalt layer (in the figure the designation – «layers»). The inset of Figure 4a,d show the $M_S$ (T) dependences measured by the Kerr method on the substrate side of samples A1 and A4 (labeled «subs» in the figure), respectively.

The dependences $M_S$ (T) measuring by a SQUID magnetometry for samples of series B are shown in the Figure 5a–d. The inset of Figure 5b shows the $M_S$ (T) dependence measured by the Kerr method measurements on the front side of the cobalt layer of samples B3.

At low temperatures a minimum of saturation magnetization appears. Measurement of Ms (T) on the face of cobalt showed the presence of a minimum at temperatures T = 120→150 K (Figure 4) for samples of series A. Measurement on the reverse side (on the substrate side) showed a minimum at temperature: T = 200→250 K. The dependences $M_S$ (T) measured by SQUID magnetometry showed the presence of a minimum at temperatures T = 40→80 K for samples of series B (Figure 5). Further, the magnetization has a peak.

Note that the position of the minimum in the $M_S$(T) dependences coincides with the position of the minimum in the I(T) dependences for samples of series B measured by

SQUID magnetometry. For samples of the A series, the average temperature of the minima of the dependence $M_s(T)$ obtained from both sides of the cobalt layer measured by the Kerr effect coincides with the minimum in the dependence $I(T)$. The difference in minimum temperatures is explained by the research method. The resonance spectrum and the SQUID magnetometry method give results from the entire sample volume. The Kerr effect method measures magnetic data from the surface layer of cobalt, since the light beam does not penetrate deep into the sample volume (20–30 nm).

The temperature dependences of the width of each Lorentzian line ($H_{WHM}$) and the resonance field ($H_{res}$) for the resonance spectra of $Al_2O_3/Ge/Al_2O_3/Co$ structures are shown in Figure 6a–d, respectively.

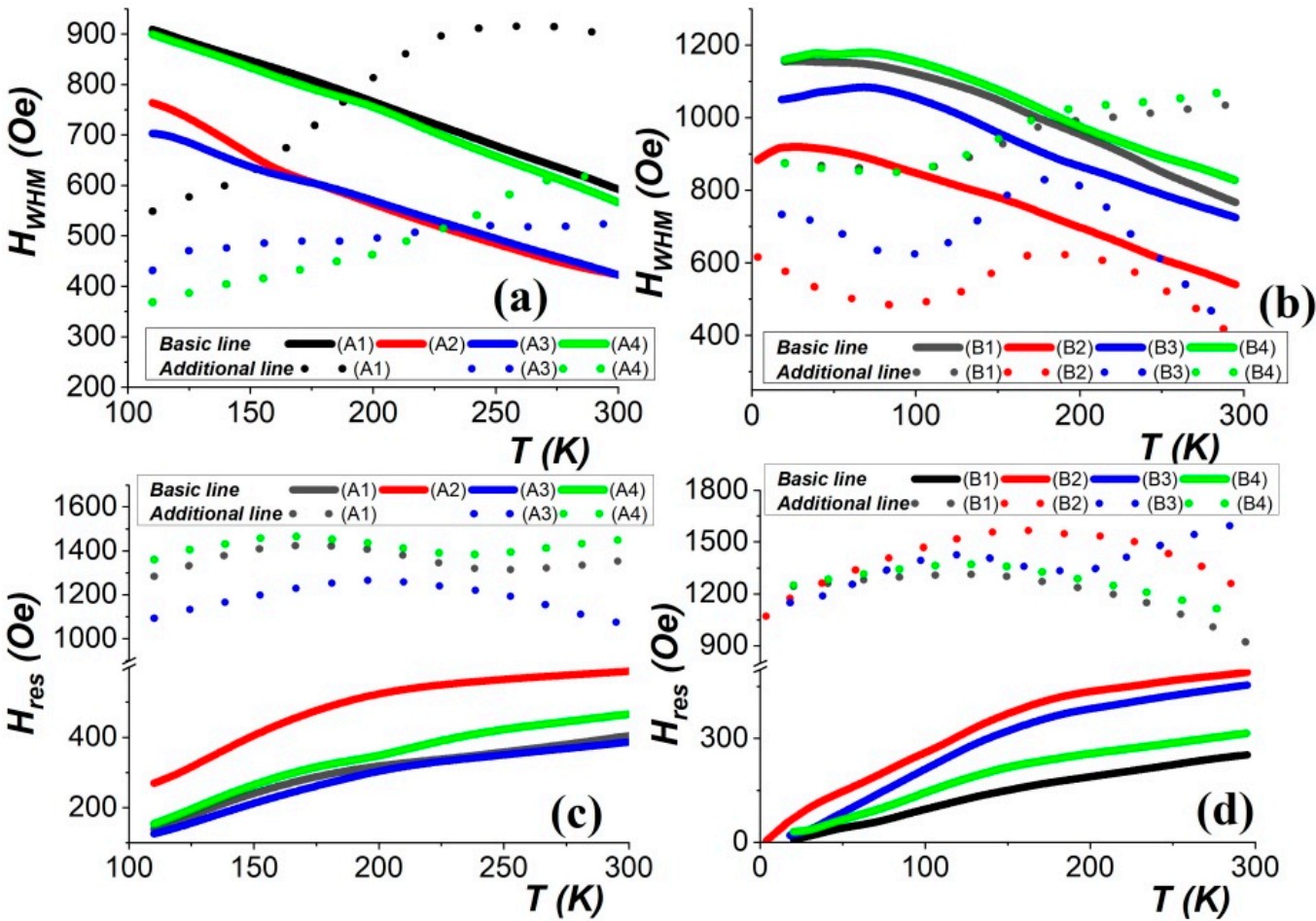

**Figure 6.** Temperature dependence of the resonance linewidth ($H_{WHM}$) of case «A» (**a**) and of case «B» (**b**), resonant field values ($H_{res}$) of case «A» (**c**) and of case «B» (**d**) of the system $Al_2O_3/Ge/Al_2O_3/Co$. Basic line: solid line. Additional line: dot line.

A distinctive feature of the temperature dependences of the line width ($H_{WHM}$) of the $Al_2O_3/Ge/Al_2O_3/Co$ system is the presence of "kinks" on the additional lines (dotted lines).

For the case "A", the width of the main lines of the resonance spectra (solid lines) decreases throughout the temperature range. In case "B" the width of the main lines of resonance spectra (solid lines) is weakly sensitive to temperature for samples B1 and B4 at temperatures T = 50→100 K. For samples B2 and B3 there is a maximum at about 50 K. A further increase in temperature up to 300 K shows a significant decrease in $H_{WHM}$.

For the case "A", the width of the additional lines of the resonance spectra (dotted lines) increases throughout the temperature interval with small "kinks" in the temperature interval T = 125→225 K. In the case of "B", the width of the additional lines of the resonance

spectra (dotted lines) has a minimum in the temperature interval T = 50→100 K, then increases up to the temperature T~180 K. With further increase in temperature up to 300 K, a significant decrease in $H_{WHM}$ is observed for samples B2 and B3 as for sample A2 and is weakly sensitive to temperature for samples B1 and B4 similar to the behavior of samples A1 and A4. A significant increase in the resonance line width observed at low temperatures indicates a strong attenuation of the magnetization precession.

The value of the resonance field Hres changed with increasing temperature. Here we can distinguish two regions for an additional decomposition line (dashed lines) in both cases of sample preparation. At temperatures of 4.2—150 K, an increase in Hres was observed. A maximum is seen in the vicinity of 150—200 K. Further there is a decrease of Hres up to 300 K. Thus, the behavior of temperature dependences of $H_{WHM}$ and Hres, both main lines and additional lines for both sputtering cases ("A" and "B"), is almost the same.

The appearance of a minimum in the M (T) curves in pure ferromagnetic cobalt at low temperatures indicates the presence of a spin glass feature [27,28] or other kinds of magnetic inhomogeneity. For example, paramagnetic particles may be present [29]. The broadening of the magnetic resonance signal of the complementary line and the simultaneous reduction of the main line may be due to the competition of the two magnetic phases (in the presence of oxygen or paramagnetic metal ions). The local field acting on a paramagnetic particle is composed of the external field H and the field Hd created by the dipoles (magnetic moments) of neighboring paramagnetic particles. The Hd field changes from point to point, because the set of neighboring paramagnetic particles and the direction of their magnetic moments change, which leads to a broadening of the resonance line.

It is known that the intensities of resonance spectra (the areas under the absorption lines of resonance spectra) are proportional to the number of unpaired electrons, i.e. to the concentration of magnetic particles in the sample. Figure 7 shows the temperature dependences of intensities (areas under lines) of each line of resonance spectra of structure $Al_2O_3/Ge/Al_2O_3/Co$.

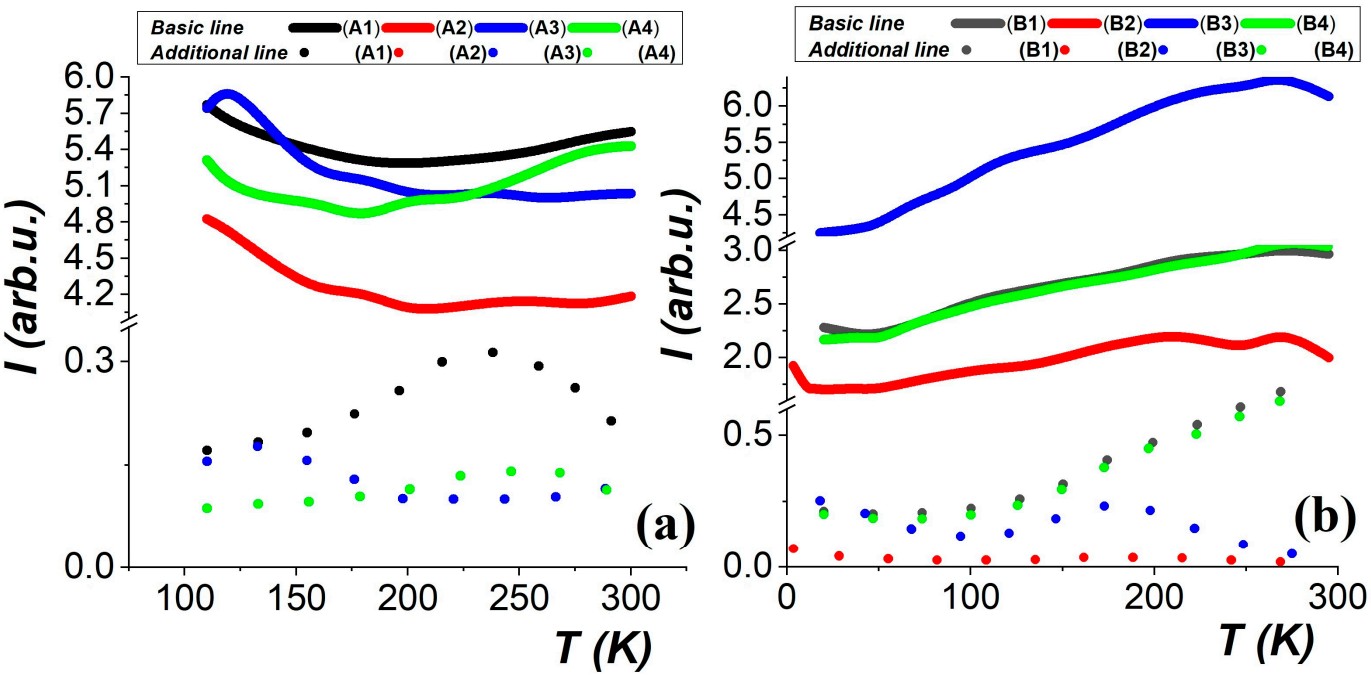

**Figure 7.** Temperature dependences of the integral intensities (areas under resonance spectra) of each line of the resonance spectrum of structure $Al_2O_3/Ge/Al_2O_3/Co$ of case "A" (**a**) of case "B" (**b**). Basic line: solid line. Additional line: dot line.

In our case, during the growth of the cobalt layer at the boundary with the aluminum oxide layer and in the cobalt volume, a large number of finely dispersed particles of both pure cobalt and cobalt-oxygen-aluminum compounds appear, as follows from the

microscopy data. This leads to the presence of an additional magnetic phase in both sputtering cases ("A" and "B"). The temperatures of the minima differ for the deposition methods. This means that the granules formed in cobalt have a distinct structure.

The ZFC/FC technique makes a significant contribution to the behavior of the microstructure and magnetic properties of granular media. It allows us to qualitatively determine the existence of superparamagnetism in the system and draw conclusions about the presence of a granular state in the object under study. Thus, in the structure of $Co_{36}Al_{22}O_{42}$ for the temperature dependence of magnetization there is a "freezing" or blocking temperature at T= 50 K. The blocking temperature for $Co_{52}Al_{20}O_{28}$ and $Co_{66}Al_{13}O_2$ structures rises to T = 150 and 160 K, respectively [30].The magnetic resonance signal from Co 2+ ions in the octahedral environment can be observed only at reduced (close to helium) temperatures. T = 2→40 K [31].

In the crystal structure of aluminum oxide (corundum) or in spinel two positions are possible, where Co 2+ ions can be located. These can be either structural aluminum positions with octahedral symmetry of the nearest surroundings, or tetrahedral positions originally vacant in corundum, but not in spinel. The temperature at which the minima for the main line are observed in the investigated structures of the "B" case is close to the helium temperature T~30 K (Figure 7b). Therefore, it is possible that this main line also decomposes into components.

It is known that the recoverability of cobalt depends on the particle size [32]. The diameter of cobalt crystallites affects the activity and selectivity of catalysts during growth. However, these data are very contradictory. Thus, enhanced cobalt-alumina particle interactions are responsible for the lower aggregation of metallic cobalt particles due to the formation of smaller cobalt particles of about 13–70 nm. This was confirmed by X-ray photoelectron spectroscopy, temperature-programmed reduction, and PEM analysis mainly [33,34].

A large proportion of the impurity phase occurs on the aluminum oxide layer. The temperature of the magnetization minimum, when it is removed from the two sides of the cobalt layer is different. The value of the temperature minimum on the side of the layers is about 150 K for case "A". As the cobalt layer grows, the granules become larger. The temperature minimum on the back side of the cobalt layer increases to a temperature of 200–250 K.

Thus, at least two magnetic phases appear as a result of the growth of the cobalt layer. One contribution is from the spins of ferromagnetic particles (the hcp of cobalt pellets), and the other contribution is from the magnetically disordered phase of finely dispersed cobalt particles and $Co-Al_2O_3$ granules, Magnetic disorder causes a decrease in the saturation magnetization. At low temperatures, these regions are spin-correlated. As the temperature increases, the magnetic field continues to hold the spin-correlated regions, with the ferromagnetic particles having a predominant influence on the magnetization of the system.

## 3. Conclusions

The technological conditions, structure, surface, magnetic and resonance properties of $Al_2O_3/Ge/Al_2O_3/Co$ films have been systematically investigated by atomic force and electron microscopy, Kerr magneto-optical effect method, SQUID magnetometry and magnetic resonance. The magnetization saturation value of $Al_2O_3/Ge/Al_2O_3/Co$ films, the magnetic resonance line width, the coercive force, and the hysteresis loop rectangularity coefficient are essentially determined by the production technology and the film texture. Let us formulate the main results of the work.

It was found that the surface roughness of the $Al_2O_3/Ge/Al_2O_3/Co$, structure increases manyfold if atmospheric air from a clean room is introduced into the working chamber before sputtering the last layer. The decrease in the sputtering rate of non-magnetic layers, as well as the decrease in the deposition rate of cobalt in the system,

leads to a decrease in the roughness and coercive force value of cobalt deposited on nonmagnetic layers.

The thermomagnetic curves revealed the appearance of minima observed at low temperatures on both sides of the cobalt layer. A study of the magnetic resonance response also showed the appearance of a minimum on the temperature dependence of the integral intensity.

The values of the temperature minima differ in magnitude on both sides of the cobalt layer. This indicates the existence of at least two magnetic phases in the cobalt layer. One contribution is from the spins of ferromagnetic particles (cobalt hcp granules), and the other is from the magnetically disordered phase of finely dispersed cobalt particles and $Co-Al_2O_3$ granules.

Sources of precession perturbations at low temperatures can be explained by: the influence of the $Al_2O_3$ layer structure on the $Al_2O_3/Co$ interface; residual gases in the working chamber atmosphere, and finely dispersed cobalt pellets distributed over the cobalt film thickness.

The following phenomena are formed as a result:

(1) Diffusion of cobalt particles and $Al_2O_3$ layer and formation of weakly magnetic interface.
(2) Influence of $Al_2O_3$ layer growth structure on cobalt layer growth and, consequently, predominance of either shape anisotropy or crystallographic magnetic anisotropy of ferromagnetic particles.
(3) The formation of $Co-Al_2O_3$ granules at the interface and in the volume of the cobalt layer.
(4) Formation of the CoO antiferromagnetic layer on the atmospheric side.

The results of this work can be useful for controlling the roughness parameters of ferromagnetic films and thereby affecting the value of Hc and $H_{WHM}$, which allows optimizing the function of spintronics and microwave devices.

**Author Contributions:** Conceptualization, A.V.K. and G.S.P.; methodology, A.V.K.; software, Y.G.S.; validation, A.V.K. and S.M.Z.; formal analysis, N.N.K.; investigation, N.N.K.; V.I.Y. and R.Y.R.; writing—review and editing, A.V.K. and N.N.K. All authors have read and agreed to the published version of the manuscript.

**Funding:** The work was carried out in the process of fulfilling the state assignment of the Ministry of Science and Higher Education of the Russian Federation № FSRZ—2020—0011 "Synthesis and physical basis of nanoscale film and granular composite materials for spintronics devices".

**Institutional Review Board Statement:** Not applicable.

**Informed Consent Statement:** Not applicable.

**Data Availability Statement:** Not applicable.

**Acknowledgments:** Sample preparation of samples to determine the morphology of structures was carried out using the equipment of the Center for Collective Use of the Federal Research Center KSC SB RAS by M.N. Volotchaev. SAED patterns were interpreted in the Laboratory of Electron Microscopy of the Joint Scientific Center of Siberian Federal University by R.R. Altunin. We really appreciated reviewers for their constructive criticism and recommendations to improve the MS.

**Conflicts of Interest:** The authors declare no conflict of interest.

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
