# Peer review of "Magnetic and Resonance Properties of a Low-Dimensional Cobalt–Aluminum Oxide–Germanium Film Tunnel Junction Deposited by Magnetron Sputtering"

_magnetochemistry, doi:10.3390/magnetochemistry8100130_

Round 1
Reviewer 1 Report
This work studies the Ge/Al2O3/Co heterostructure. The authors did sufficient experiments to characterize its interface. However, the authors must clarify the following comments and fix this paper seriously before it can be published.
1) The writing of this article has serious problems. The title should not have a punctuation. The last of the abstract lacks of a full stop. The short name should be uniform. Show readers its full name in the first time, such EPR. And pay attentions for the subscripts such MS. The whole paper was written half-heartedly. There are a lot of mistakes in grammar, such as Line 24-26 and Line 270.
2) The figures in a paper should be orderly and unabridged. The subfigures should be aligned. The authors should revise and reprepare all of figures.
3) In Page 1, the authors should introduce the background concisely in Abstract.
4) In Page 1-2, the introduction should be transferred to the main work concisely. The authors should shorten the introduction.
5) In Page 4, the TEM and EDS in Figure 1 should be rotated parallelly for comparing.
6) In Page 4 (Line 149), “Fig. 1a” should be “Fig. 2a”. Please check the order of figures.
7) In Page 5, the authors should present the physical quantity not “A” and “B” in Figure 3.
8) In Page 6 (Line 198-202), the authors should explain their claim in details or cite some references.
9) The authors should cite some related work such as iScience 23(2020)101614.
Author Response
- The writing of this article has serious problems. The title should not have a punctuation. The last of the abstract lacks of a full stop. The short name should be uniform. Show readers its full name in the first time, such EPR. And pay attentions for the subscripts such MS. The whole paper was written half-heartedly.There are a lot of mistakes in grammar, such as Line 24-26 and Line 270.
Answer: we agree. Corrected.
2) The figures in a paper should be orderly and unabridged. The subfigures should be aligned. The authors should revise and reprepare all of figures.
Answer: we agree. Corrected.
- In Page 1, the authors should introduce the background concisely in Abstract.
Answer: we agree. Corrected.
4) In Page 1-2, the introduction should be transferred to the main work concisely. The authors should shorten the introduction.
Answer: we agree. Corrected.
5) In Page 4, the TEM and EDS in Figure 1 should be rotated parallelly for comparing.
Answer: we agree. Corrected.
6) In Page 4 (Line 149), “Fig. 1a” should be “Fig. 2a”. Please check the order of figures.
Answer: we agree. Corrected.
7) In Page 5, the authors should present the physical quantity not “A” and “B” in Figure 3.
Answer: we agree. Corrected.
8) In Page 6 (Line 198-202), the authors should explain their claim in details or cite some references.
Answer: we agree. Corrected.
9) The authors should cite some related work such as iScience 23(2020)101614.
Answer: we agree. Corrected.

Reviewer 2 Report
The article "Magnetic and resonant properties of the low-dimensional tunneling transition of cobalt-Al2O3-germanium" deals with the methods of sample preparation and study of their magnetic properties. Very good quality samples were synthesized.
In my opinion, the paper is well written. But, some revisions are necessary to be enumerated as follows:
1) In the article, the word “FMR” is better to replace to “magnetic resonance”
2) Line 149 in text - “Fig 2a” should be instead of “Fig 1a”
3) Figure 2 is rather bulky. I advise to divide it into two separate Figures for samples A and B. What do “B1_1 and B2_1 … ” mean (Fig 2d)? You’d better add information about it, please.
4) There is an error in the caption for Figure 2. It’s necessary to replace "for samples A3, A4" to " for samples A1, A4" (line 160).
5) Figure 3. No units of measurement.
6) Where is the caption for Figure 4? You’d rather add a more detailed description of the curves to the caption of Figure 4. The solid line is the main line, and dotted – additional line. What do “A1_1” and “A1_2” mean? You might add information to the caption for the Figure.
7) In the Figure 4 try to replace "HWHM" to "HWHM". There is confusion in data in Figure 4:
- A3_2 is a dott line, then A3_2 again (Fig 4a).
- And red dott line is A1_2 on Fig 4c?
- Also in Figure 4 (Fig. 4 (c,d)), there is confusion in the data for curves B, they aren’t done in the order.
- You have to correct this Figure.
8) The part of the article after Figure 4 is missed (Line 181)? Please, check this.
9) There isn’t the caption for Figure 5?
In general, the manuscript has a correct methodological structure. I hope that authors reconsider these all points and they clarify the pointed issues.

Author Response
- In the article, the word “FMR” is better to replace to “magnetic resonance”
Answer: we agree. Corrected.
- Line 149 in text - “Fig 2a” should be instead of “Fig 1a”
Answer: we agree. Corrected.
- Figure 2 is rather bulky. I advise to divide it into two separate Figures for samples A and B. What do “B1_1 and B2_1 … ” mean (Fig 2d)? You’d better add information about it, please.
Answer: we agree. Corrected.
- There is an error in the caption for Figure 2. It’s necessary to replace "for samples A3, A4" to " for samples A1, A4" (line 160).
Answer: we agree. Corrected.
- Figure 3. No units of measurement.
Answer: we agree. Corrected.
- Where is the caption for Figure 4? You’d rather add a more detailed description of the curves to the caption of Figure 4. The solid line is the main line, and dotted – additional line. What do “A1_1” and “A1_2” mean? You might add information to the caption for the Figure.
Answer: we agree. Corrected.
7) In the Figure 4 try to replace "HWHM" to "HWHM". There is confusion in data in Figure 4:
- A3_2 is a dott line, then A3_2 again (Fig 4a).
- And red dott line is A1_2 on Fig 4c?
- Also in Figure 4 (Fig. 4 (c,d)), there is confusion in the data for curves B, they aren’t done in the order.
- You have to correct this Figure.
Answer: we agree. Corrected.
- The part of the article after Figure 4 is missed (Line 181)? Please, check this.
Answer: we agree. Corrected.
- There isn’t the caption for Figure 5?
Answer: we agree. Corrected.

Reviewer 3 Report
The authors studied the magnetic and resonance properties of Al2O3/Ge/Al2O3/Co films by the magnetization measurement and the EPR response. The structural analysis and the discussion of surface morphology need to be strengthened. The discussion of magnetism also needs more description. This paper should be reconsidered after major revision.
1. The XRD pattern should be provide in this work.
2. The SEM (and particle distribution) and EDS (or EDX) data should be described in this paper.
3. Discussions of relevant literature on structure, SEM and EDS could be further enhanced, which can link to the existing work. Authors might consider the following relevant recent work in this regard: https://doi.org/10.1063/5.0078188.
4. When the abbreviation appears for the first time, the full name should be indicated, such as EPR.
5. The deposition rate is mentioned in table 1. However, what is the thickness of each layer in Al2O3/Ge/Al2O3/Co films? These should be mentioned in this paper.
6. There are a lot of magnetic data in Figure 2, but there is very little analysis and discussion on magnetic data. The discussion of the data in Figure 2 needs more description.
7. What does the X-axis and Y-axis in Figure 3 mean? Please modify Figure 3.
8. The paper is not written well. The authors can refer to the literature mentioned above [https://doi.org/10.1063/5.0078188] to learn how to write a paper. After improving all the above issues, the revised paper can be reconsidered.
Author Response
- TheXRD pattern should be provide in this work.
Answer: we agree. We added new images.
- TheSEM (and particle distribution) and EDS (or EDX) data should be described in this
Answer: we agree. We added new images.
- Discussions of relevant literature on structure, SEM and EDS could be further enhanced, which can link to the existing work. Authors might consider the following relevant recent work in this regard: https://doi.org/10.1063/5.0078188.
- When the abbreviation appears for the first time, the full name should be indicated, such as EPR.
Answer: we agree. Corrected.
- The deposition rate is mentioned in table 1. However, what is the thickness of each layer in Al2O3/Ge/Al2O3/Co films? These should be mentioned in this paper.
Answer: we agree. Corrected.
- There are a lot of magnetic data in Figure 2, but there is very little analysis and discussion on magnetic data. The discussion of the data in Figure 2 needs more description.
Answer: we agree. Corrected.
- What does the X-axis and Y-axis in Figure 3 mean? Please modify Figure 3.
Answer: we agree. Corrected.
- The paper is not written well. The authors can refer to the literature mentioned above [https://doi.org/10.1063/5.0078188] to learn how to write a paper. After improving all the above issues, the revised paper can be reconsidered.
Answer: Thanks for the offer. We used the link when correcting mistakes and remarks.

Round 2
Reviewer 1 Report
The authors have carefully revised the paper according my previous comments. I am happy with the current format of the paper.
Reviewer 3 Report
The paper has been improved. Magnetic and resonance properties can be investigated by the combination of Magnetization Measurement and the EPR Measurement. But the kind of characterization technology has been reported in previous work on other magnetic materials [Hui Zhang, Yan Wang, Haiou Wang, Dexuan Huo, and Weishi Tan, Room-temperature magnetoresistive and magnetocaloric effect in La1−xBaxMnO3 compounds: Role of Griffiths phase with ferromagnetic metal cluster above Curie Temperature, Journal of Applied Physics 131, 043901 (2022)]. I still think that the reference could be cited in the introduction or discussion part.
Author Response
We agree with the remark. Corrected intro:
Among the methods for studying the magnetic and resonant properties of ferromagnetic metal particles or clusters, one can single out methods for combining measurements of magnetization and measurements of electron paramagnetic resonance. For example, such methods have yielded results in the study of the magnetoresistive and magnetocaloric effect in La1−xBaxMnO3 compounds: for explaining Role of Griffiths phase with ferromagnetic metal cluster above Curie temperature [25].
We would like, if it is still possible, to supplement the title of the manuscript:
Magnetic and resonance properties of a low-dimensional cobalt - aluminum oxide - germanium film tunnel junction deposited by magnetron sputtering
